# The hierarchy of sugar catabolization in *Lactococcus cremoris*

Sieze Douwenga,[1,2] Berdien van Olst,[1,3,4] Sjef Boeren,[1,4] Yanzhang Luo,[5] Xin Lai,[2] Bas Teusink,[1,2] Jacques Vervoort,[1,4] Michiel Kleerebezem,[1,3] Herwig Bachmann[1,2,6]

**ABSTRACT** Bacteria adapt to nutrient availability by regulating the synthesis of enzymes. Transcriptome- and multi-sugar growth studies suggest that *Lactococcus cremoris* represses genes involved in the catabolization of lower growth rate-supporting (lower quality) sugars in a hierarchical order. Furthermore, *L. cremoris* appears to always express genes involved in the catabolization of higher growth rate-supporting sugars (higher quality) relative to the sugar it is growing on. Here, we unraveled the sugar catabolization hierarchy by determining the sugar catabolizing capacity and the proteome of cells exponentially growing on glucose ($\mu_{max}$ = 0.72 h$^{-1}$), lactose ($\mu_{max}$ = 0.6 h$^{-1}$), galactose ($\mu_{max}$ = 0.44 h$^{-1}$), or maltose ($\mu_{max}$ = 0.43 h$^{-1}$). We found that *L. cremoris* can grow on 14 of the 96 sugars in a Biolog plate, with $\mu_{max}$ ranging from 0.32 to 0.72 h$^{-1}$. Proteome and catabolization rate measurements show that *L. cremoris* consistently prepares for the catabolization of higher-quality sugars, except trehalose. While cells were not prepared for the catabolization of most lower-quality sugars, some proteins related to fructose and lactose consumption were always present. Moreover, reducing the growth rate of glucose through salt stress had only a minor influence on the sugars that *L. cremoris* could catabolize. These findings demonstrate that the catabolization hierarchy is not strictly linked to absolute growth rate or sugar quality. Cells instantly catabolizing a higher-quality sugar require enhanced expression of ribosomal and nucleotide metabolism functions for growth rate maximization, whereas transitioning to lower-quality sugars requires enhanced synthesis of proteins related to arginine catabolism and mixed acid fermentation, besides sugar-specific catabolic proteins.

**IMPORTANCE** The availability of nutrients to microorganisms varies considerably between different environments, and changes can occur rapidly. As a general rule, a fast growth rate—typically growth on glucose—is associated with the repression of other carbohydrate utilization genes, but it is not clear to what extent catabolite repression is exerted by other sugars. We investigated the hierarchy of sugar utilization after substrate transitions in *Lactococcus cremoris*. For this, we determined the proteome and carbohydrate utilization capacity after growth on different sugars. The results show that the preparedness of cells for the utilization of "slower" sugars is not strictly determined by the growth rate. The data point to individual proteins relevant for various sugar transitions and suggest that the evolutionary history of the organism might be responsible for deviations from a strictly growth rate-related sugar catabolization hierarchy.

**KEYWORDS** *Lactococcus*, catabolism, catabolite repression, proteome, sugar transition

Address correspondence to Herwig Bachmann, h.bachmann@vu.nl.

Sieze Douwenga and Berdien van Olst contributed equally to this article. Authors are ordered alphabetically based on the last name.

Jacques Vervoort is deceased.

The project is organized and executed under the auspices of TiFN, a public-private partnership on precompetitive research in food and nutrition. H.B. is part-time employed by NIZO Food Research. The authors have declared that no competing interests exist in the writing of this publication. Funding for this research was obtained from Friesland Campina (Wageningen, The Netherlands), CSK Food Enrichment (Wageningen, The Netherlands), and Top-sector Agri&Food.

To thrive in a variable environment, bacteria have to continuously adapt and maximize their fitness. This is typically achieved by regulating protein synthesis and activity. To detect sugars that are available in the environment, microorganisms often synthesize basal levels of the corresponding transporter and catabolic proteins. If a sugar

appears, a sugar-specific intermediate activates the full synthesis of proteins involved in its catabolization. Conversely, basal synthesis of these proteins can be fully repressed through carbon catabolite repression (CCR) systems, of which the mechanisms vary between organisms (1). Full repression can be exerted for a sugar that supports a lower growth rate than the current growth rate. In multi-sugar environments, this regulation often results in the sequential consumption of certain sugars, i.e., diauxic growth (2–4). CCR during growth on glucose has been studied in detail for many microorganisms. Typically, high growth rates on excess glucose come with enhanced repression of genes involved in the catabolization of other carbon sources, whereas growth-limiting glucose concentrations alleviate such repression (5, 6).

Several studies have also investigated the repression of gene expression during growth on sugars other than glucose. In *Escherichia coli*, the synthesis of sugar-metabolizing proteins follows a hierarchy that highly corresponds with the sugar quality, i.e., the maximum achievable growth rate on a sugar (7). In *Bacillus subtilis*, the repression strength for growing on different sugars also followed a clear hierarchy, which could be explained by CCR-mediated regulation of protein synthesis but did not strictly correlate with sugar quality (8).

The mechanism underlying CCR varies between bacteria, but in firmicutes like *B. subtilis* and lactic acid bacteria, catabolite repression is regulated in part by the Catabolite control protein A (CcpA). CcpA regulates gene expression through the formation of a complex with serine 46-phosphorylated histidine-containing phosphocarrier protein (HPr-Ser46-P). The amount of phosphorylated HPr-Ser46 differs between various sugars, and hence the degree of CCR-mediated gene regulation differs between various sugars (8). HPr-Ser46 phosphorylation is triggered by high concentrations of fructose 1,6-bisphosphate, glucose-6-phosphate, and ATP and low concentrations of free phosphate (1). These concentrations coincide with a high glycolytic flux, which commonly coincides with high growth rates associated with the availability of higher-quality sugars (1). A general repression hierarchy can then be realized through a varying CcpA-Hpr-Ser46-P complex binding affinity for the *cre* sites on DNA (transcription is repressed when this complex binds), which are linked to operons encoding sugar-specific catabolic proteins. Additionally, when HPr is phosphorylated at the His15 residue, it phosphorylates the glycerol kinase GlpK and transcriptional regulators, which are required for the activation of catabolic operons. Notably, HPr-His15 phosphorylation occurs at high PEP concentrations (coinciding with low glycolytic flux) (9). Furthermore, CcpA-Hpr-Ser46-P-mediated repression mostly functions through transcription inactivation. Additionally, this CcpA complex activates transcription of the las operon, which encodes glycolytic enzymes that are linked to high growth rates (phosphofructokinase, pyruvate kinase, and L-lactate dehydrogenase) (10).

Given the mechanisms that govern CCR, one could conclude that bacteria only synthesize proteins for sugars that support a higher growth rate. However, in *B. subtilis*, a direct effect of growth rate on the hierarchy of sugar catabolization was deemed unlikely (8). Accordingly, in *E. coli*, the hierarchy can also not be fully explained by sugar quality or growth rate since *E. coli* shows much weaker repression when grown on arabinose than on glucose, despite a comparable growth rate on both sugars (5). Furthermore, *E. coli* that was grown alternatingly on maltose or lactose expressed low levels of maltose catabolic genes when growing on lactose and was shown to lose expression of these genes following 500 generations of growth on lactose. This shows that the evolutionary history of an organism can rapidly result in deviations from a sugar quality-based repression hierarchy (11).

While many lactococci have been isolated from dairy environments, they can also be found on plant material. Several studies suggest that the adaptation of plant isolates to growth in a dairy environment led to the loss of genes that were not required for growth in milk, while the evolution of the proteolytic system enhanced the strain's capacity to utilize milk proteins (12–14). It is likely that repression and activation of genes related to

sugar catabolization also changed in the dairy environment due to the excess presence of lactose in dairy environments compared to the various sugars in plant environments.

The presence of a sugar catabolization hierarchy (i.e., the preference an organism has for the consumption of different sugars) in *Lactococcus cremoris* (formerly *Lactococcus lactis* subsp. *cremoris*) (15) is suggested in several studies involving multi-sugar environments and CCR. In the presence of glucose, galactose, and lactose, *L. lactis* $ML_3$ first co-consumes glucose and lactose and only consumes galactose at low glucose and lactose concentrations (16). In addition, adding glucose or lactose to galactose-grown *L. lactis* $ML_3$ cells results in an instantaneous switch to higher growth rates on the added sugar, indicating that these cells synthesize glucose and lactose catabolization enzymes even in the absence of these sugars (16). In line with this, glucose and lactose were preferred in glucose-maltose and lactose-maltose-grown *L. lactis* 65.1, and maltose was only consumed after a lag phase (diauxic growth), whereas maltose and galactose were co-consumed (17). Furthermore, when transitioning *L. cremoris* MG1363 from a glucose to a cellobiose environment, only a subpopulation of cells was found to continue growth, and the size of that cellobiose utilizing subpopulation decreased when the glucose was depleted faster (18). A similar study in yeast showed that the lag phase for continued maltose growth is directly proportional to the glucose concentration at which maltose catabolic genes are activated, and it varies between different yeast isolates (6). The variation was shown to depend on the alternating frequency and duration of glucose and maltose presence (6). Hence, we hypothesize that in lactococci, de-repression occurs in a similar fashion not only for glucose-to-cellobiose transitions but also for transitions between other high- and low-growth rate-supporting sugars. As mentioned earlier, the mechanisms of CCR suggest that this is likely to be based on the current growth rate and/or glycolytic flux. Biomass yield on sugar might be considered an alternative measure of sugar quality. However, in environments where resources are shared (like the dairy environment), faster-growing cells will outcompete cells with a lower growth rate and a high yield (19). This suggests that growth rate is a major determinant of CCR.

The strength with which the CcpA protein represses different genes related to sugar catabolization in *L. cremoris* MG1363 also appears to follow a nutrient quality hierarchy. This is based on a transcriptome comparison of wild-type *L. cremoris* MG1363 with a *ccpA* gene deletion mutant, both grown on glucose (20). In this study, there was a clear order of CcpA repression strength for genes related to the consumption of different sugars. However, the absence of CcpA in this mutant also means that its concentration cannot be fine-tuned and its activity cannot be modulated by the formation of a complex with different levels of Hpr-Ser46-P (which is controlled by the glycolytic flux intermediates). Moreover, the CcpA repression strength for different sugar catabolization genes was only determined during growth on glucose. Therefore, a sugar catabolization hierarchy based on the transcriptome study by Zomer et al. may not reflect the true hierarchy for wild-type *L. cremoris* (20).

Whereas the studies mentioned above investigated the expression levels of sugar metabolizing pathways in different environments, or lag-phases in multi-sugar environments (16–18, 20), neither the sugar catabolization hierarchy nor the role of sugar type in the expression of other sugar catabolizing pathways is known for *L. cremoris*. It is also not known whether this hierarchy follows sugar quality for a large set of sugars in a single strain.

We here aim to characterize the sugar catabolization hierarchy of *L. cremoris* NCDO712 (21) by determining which sugars it could utilize when precultured at four carbon-source determined growth rates, ranging from 0.45 to 0.72 $h^{-1}$. Characterizations included the full proteome as well as the rate of catabolization of 12 of the growth-supporting sugars in a Biolog plate. The results largely support the hypothesis that the sugar catabolization hierarchy in *L. cremoris* is based on synthesizing proteins related to the consumption of higher-quality sugars but not lower-quality sugars. However, the results also illustrate exceptions (e.g., high-quality sugar trehalose is not prepared for), showing

that *L. cremoris* sometimes deviates from a sugar quality-based sugar catabolization hierarchy.

## MATERIALS AND METHODS

### Strains and media

Strains were cultured in a chemically defined medium for prolonged cultivation (CDMpc) at 30°C, as described earlier (22). The carbon source and its concentration are described separately with each experiment.

The strains used in this study were *L. cremoris* NCDO712 and *L. cremoris* MG1363. *L. cremoris* NCDO712 carries six plasmids (21), whereas *L. cremoris* MG1363 is a plasmid-free and phage-cured derivative of *L. cremoris* NCDO712 (23). Before use, *L. cremoris* NCDO712 was grown in lactose CDMpc. To this end, M17 broth (Oxoid #CM0817) + 2.5% (wt/vol) lactose was inoculated from a −80°C glycerol stock, grown until stationary phase ($OD_{600}$ = 3.05), and subsequently plated on 12.5 mM lactose CDMpc agarose plates. A single colony was transferred to 7.5 mM lactose CDMpc and grown until the stationary phase of growth ($OD_{600}$ = 0.73). It was subsequently serially propagated by making 1:100 dilutions five times in 12.5 mM lactose CDMpc (average stationary phase $OD_{600}$ = 0.81). Finally, the culture was plated on a 12.5 mM lactose CDMpc agarose plate, and a single colony was picked. This was grown until the stationary phase of growth in CDMpc supplemented with 3.75 mM lactose ($OD_{600}$ = 0.58), after which glycerol was added to a concentration of 20% (wt/vol). The low concentration of lactose prevented acidification of the medium, potentially reducing stress during storage. The culture was then aliquoted into 300 µL single-use vials and stored at −80°C until further use.

A similar procedure was carried out for *L. cremoris* MG1363, except the preculture in M17 was carried out with 2.5% (wt/vol) glucose. Additionally, in CDMpc, the glucose concentration was always 25 mM; this includes the final culture before aliquoting.

### Batch bioreactor cultivation

Batch cultivation took place in self-made 0.55 L bioreactors with 0.5 L working volume (22), under continuous stirring with magnetic stirrers, while standing in 30°C water baths. The headspace was continuously sparged with 5% $CO_2$ + 95% $N_2$ gas. Batch growth experiments were carried out in quadruplicate in CDMpc with the following variables: 5 mM galactose, 12.5 mM maltose, 12.5 mM lactose, 25 mM glucose, or 25 mM glucose + 500 mM NaCl.

To start up a batch cultivation, the medium was inoculated with *L. cremoris* NCDO712 and grown at 30°C for 20–30 generations in CDMpc with the same composition as that in the bioreactor. At the end of the preculture, cells were ensured to be midlog and had been dividing for eight generations (through periodic $OD_{600}$ measurements) before inoculation in the bioreactor.

Time series and single-time point samples were taken aseptically from each bioreactor.

Samples of 1 mL were periodically taken for optical density measurements ($OD_{600}$), which were performed with an external spectrophotometer (Ultrospec 2100 Pro, Amersham Biosciences) at 600 nm.

### Nuclear magnetic resonance spectroscopy

Nuclear magnetic resonance (NMR) spectroscopy measurements were done to determine the ratio of acetate over lactate production to compare homolactic acid versus mixed acid fermentation, while the amounts of ornithine produced were measured as a proxy for the arginine deiminase pathway and expressed relative to the $OD_{600}$. To this end, 2 mL of samples was taken periodically and centrifuged at 27,237 ×g for 3 min at 4°C. The supernatant was filtered through a 0.22-µm polyether sulfone filter (VWR International) and stored at −20°C until further analysis. For NMR spectroscopy,

samples were diluted 1:1 in a 10% $D_2O$/90% $H_2O$ solution. Subsequently, 1D Nuclear Overhauser Effect Spectroscopy spectra were measured on a 600 MHz NMR spectroscopy (Bruker BioSpin) equipped with a 5 mm cryorobe at a temperature of 300 K. The $^1$H-$^1$H mixing time was 10 ms. A saturation of 50 Hz was applied to water during relaxation delay and mixing. Quantum mechanics-based 1 h full spin analysis with the Cosmic Truth software (NMR Solutions Limited) was employed for data analysis and peak assignments [for details, see references (24, 25)]. Peak intensities of acetate lactate and ornithine were normalized with the total peak intensity of all peaks. Subsequently, linear regression was used to determine the ratio of the normalized peak intensity to $OD_{600}$ during the loglinear part of growth. Finally, these values were used to determine the acetate/lactate ratio and the ornithine produced relative to that produced by maltose-grown cells.

## Catabolization capacity determination using Biolog plates

Nutrient catabolization capacity was determined in exponentially growing *L. cremoris* NCDO712 cells using Biolog plates, type PM1 MicroPlate. For Biolog samples, 20–50 mL samples were taken, depending on the $OD_{600}$ of the culture at the time. Protein translation was blocked at the moment of sampling by adding erythromycin and chloramphenicol (dissolved at 1.25 mg/mL in 100% ethanol) to a concentration of 25 mg/L. Samples were then centrifuged at 4,754 ×$g$ for 15 min at 30°C. The supernatant was removed, and the pellet was resuspended in CDMpc-No carbon + 25 mg/L erythromycin and chloramphenicol at 30°C. This step was repeated once more, and the final resuspension volume was aimed at reaching an $OD_{600}$ between 0.1 and 0.3. The $OD_{600}$ of this suspension was then measured after filling a Biolog plate with 100 µL/well cells suspended in CDMpc-No Carbon + 25 mg/L erythromycin + chloramphenicol. The plate was mixed for 5 min on a plate shaker at 900 RPM at 30°C. The plate was then placed in a plate reader at 30°C, and the optical density at 460 nm ($OD_{460}$) was determined over the first 1.15 h.

The cells were translationally blocked in the PM1 MicroPlate plate to prevent adaptation to the new sugars (5). In PM1 MicroPlate plates, *L. cremoris* catabolizes (i.e., oxidizes) a sugar by simultaneously reducing tetrazolium violet to formazan (using NADH oxidase), which remains in the cell membrane and was measured at $OD_{460}$ (26). The catabolization rate was then corrected for the concentration of the translationally blocked cells by dividing by the $OD_{600}$ (measured separately using an Ultraspec 2100 pro spectrophotometer, Amersham Biosciences, before adding cells to the plate).

We checked whether any of the wells in an unused Biolog plate contained glucose using glucose strips (10–500 mg/L, MQuant).

## Proteome isolation and identification

Two milliliters of cell samples from three reactors was collected in low-protein-binding tubes and centrifuged at 27,237 ×$g$ for 3 min at 4°C. The supernatant was removed, and the pellet was frozen in liquid $N_2$ prior to storage at −20°C until further isolation. Proteomes were measured as described previously (27).

In short, frozen pellets were dissolved in 100 µL 100 mM TRIS-HCl pH 8, and the cells were lysed using a needle sonicator for 30 s with ice-cooling in between. All cell lysate proteins (including cell membranes) were reduced (15 mM Dithiotreitol, 45°C, 30 min), alkylated under denaturing conditions (20 mM acrylamide in 8 M urea in 100 mM TRIS, 10 min), loaded on a 3K Pall omega filter (12,000 rpm, 40 min), washed (130 µL 50 mM ammonium bicarbonate), and digested overnight (5 ng trypsin in 100 µL ammonium bicarbonate while shaking). Digested peptides were collected (12,000 rpm, 30 min), and the filter was washed (100 µL 1 mL/L formic acid in water). The pH of the collected sample was decreased to pH 3 using 10% trifluoroacetic acid, and the sample was stored at −20°C until measured by nanoscale liquid chromatography coupled to tandem mass spectrometry with a Proxeon EASY nLC1000 and an LTQ-Orbitrap XL mass.

The raw data files containing the MSMS spectra were analyzed by MaxQuant (version 1.6.1.0) using the *L. cremoris* MG1363 database supplemented with NCDO712

plasmid proteins and frequently observed contaminants as a reference (28). To analyze the relative abundance of proteins, their normalized label-free quantification (LFQ) intensities were used (29). The mass spectrometry proteomics data have been deposited with the ProteomeXchange Consortium via the PRIDE partner repository with the data set identifier PXD037169.

## Maximum growth rate determination of Biolog plate carbon sources

The maximum growth rate of *L. cremoris* NCDO712 and MG1363 was determined in CDMpc with one of the following carbon sources present at 0.45% (wt/vol): glucose, mannose, fructose, lactose, maltose, galactose, maltotriose, trehalose, lactulose, n-acetyl-d-glucosamine, n-acetyl-beta-d-mannosamine, mannitol, and gluconic acid. To determine the growth rate, −80°C *L. cremoris* NCDO712 stock was first inoculated into each respective medium at 0.2% (vol/vol) with a carbon source concentration of 0.09% (wt/vol) (a lower concentration was used to ensure starvation rather than acidification-related growth arrest during preculturing), which was divided over a 384-well microtiter plate (Greiner Bio-one) (20 wells per carbon source, 120 µL per well). The plate was covered with a lid and placed in a plate reader at 30°C, where cells were grown until stationary while $OD_{600}$ was measured every 2 min. Then, 0.7% (vol/vol) broth from each well was transferred to fresh medium in a new 384-well plate, this time with a 0.45% (wt/vol) carbon source, and cultivation and $OD_{600}$ measurements were again performed in a plate reader until the cells were stationary.

## Data analysis

Data analysis of $OD_{460}$ (Biolog) and $OD_{600}$ data was done using Python v3.7.4. Packages used were pandas v1.0.3 (30), numpy v1.18.1 (31), matplotlib v3.1.2 (32), seaborn v0.10.0 (33), and scipy v1.4.1 (34).

In all cases where $OD_{600}$ was measured on microtiter plates, prior to further data analysis, the background $OD_{600}$ was subtracted. The background was determined from wells filled with sterile CDMpc.

Continuous growth rates [$μ(t)$] were determined using a Savitsky-Golay filter (35) on the natural logarithm of the $OD_{600}$ data versus time. The filter was set to fit a first-order polynomial over a measurement window of 2 h, returning the first derivative as the growth rate. The maximal growth rate was determined from the continuous growth rate where $OD_{600}$ >0.018.

Carbon source catabolization rates in Biolog plates were determined from the rate of $OD_{460}$ increase over the first 22.5 min. After this time, the rate started to decline. This rate was determined by getting the first derivative from a first-order polynomial Savitsky-Golay filter on the $OD_{460}$ data with a window of 22.5 min. The rate was divided by the $OD_{600}$ of the culture, as measured in an external spectrophotometer, before the culture was divided over the Biolog plate. In the case of lactose-pregrown cells, two biological replicates were measured in the Biolog plates. For the remaining preBiolog plate growth sugars (glucose, galactose, and maltose), four replicates were measured, although some outlying catabolization rates of single sugars (single wells) in the Biolog plate were excluded, resulting in three replicates in these cases. Outliers were set to the mean value of the remaining replicates, with outliers defined as being 1.5 times outside the interquartile range. The data were then normalized by dividing all $OD_{600}$-specific catabolization rates by the catabolization rate of the sugar that the cells were previously growing on. Normalized values smaller than 0 and larger than 1 were set to 0 and 1, respectively. The background activity is an average of activities measured on several nongrowth-supporting sugars. Therefore, this cannot be subtracted from the activities in the other wells. Instead, it gives an indication of the level below which no clear conclusions can be drawn about the activity level.

Clustering of the carbon source $OD_{600}$-specific normalized catabolization rates was done using the Seaborn clustermap function with the Ward clustering algorithm and Euclidian distance metric.

Proteome comparison analysis was performed in Perseus (version 1.6.2.1). Protein groups (containing one or more proteins) that were only identified by a modified site, matched in the reverse database, contained less than two peptides, did not have a unique peptide, or did not have at least two valid LFQ measurements in at least one sugar group were removed. LFQ values were log10 transformed, and missing values were imputed from a normal distribution using the standard setting in Perseus. Significantly different synthesized proteins were determined using pairwise Student's $t$-tests with a false discovery rate of 0.05 and an $S_0$ of 0.4 (36).

The proteins that were associated with growth rate changes were identified using redundancy analysis. A redundancy analysis (RDA) is comparable to a principal component analysis (unconstrained analysis), except that the explanatory variable (first axis) is constrained to be the growth rate in this case. The analysis was performed by the vegan package (version 2.5.7) in R (version 4.0.2) on the imputed LFQ values. To identify which gene functions are enriched, gene set enrichment analysis was performed on the RDA-related response variables using the clusterprofiler package (version 3.18.1) in R. Gene functions were categorized based on their membership in the second-level Brite hierarchy. Enrichment was assumed at a $P$-value of <0.05.

## Proteome investment

Proteome investment is defined as the percentage of the proteome that needs to be newly produced to fully adapt to a new sugar ("to" sugar) during a transition. To this extent, we calculated the difference in iBAQ fraction (i.e., iBAQ values normalized for the summed iBAQ per sample) for each individual protein between any combination of two samples (132 comparisons). The positive differences indicate that the protein level is higher in cultures grown on the "to" sugar than on the "from" sugar and are summed to obtain the fraction of the proteome that must be newly produced. These fractions are averaged across the replicates per from-to combination and normalized against the "from" sugar by subtracting the summed fraction of the self-transition (i.e., the "from" and "to" sugar are identical) to correct for biological variation.

## RESULTS

### *L. cremoris* NCDO712 can grow on 14 out of the 96 sugars on the Biolog plate

We here elucidate the sugar catabolization hierarchy (i.e., the preference an organism has for the consumption of different sugars) and to what extent this is based on the sugar quality or the absolute growth rate. The sugar quality is defined as the maximum achievable growth rate of the strain in question. The absolute growth rate may be altered by sources other than the sugar quality and concentration (e.g., presence of salt stress, weak acid stress, and amino acid composition). We define the hierarchy as the extent to which *L. cremoris* NCDO712 can use sugars when transitioned between different quality sugars (and hence at different growth rates). The results were based on the normalized catabolization rates of individual sugars and the related catabolic protein abundances. We first determined which sugars NCDO712 could grow by measuring the final $OD_{600}$ after 30 h of growth on a Biolog plate. Fourteen sugars resulted in an $OD_{600}$ >0.2, which we considered to be growth-supporting (Fig. S1). Subsequently, we determined the maximal growth rates on these 14 sugars (Fig. S1 and S2), which resulted in a broad range of maximal growth rates ranging from 0.32 (gluconic acid) to 0.72 $h^{-1}$ (glucose).

We next determined if the CcpA repression strength order described by Zomer et al. for *L. cremoris* MG1363 followed the sugar quality (20). For this, we determined the maximal growth rates of the plasmid-cured NCDO712 derivative MG1363 on 12 of the 14 sugars. Lactose and lactulose were excluded as *L. cremoris* MG1363 does not have the plasmid-encoded *lac*-operon required for lactose utilization, and lactulose catabolization was never detected. While the maximal growth rate on nearly every sugar is slightly higher for *L. cremoris* MG1363 than for NCDO712, the sugar quality hierarchy (i.e., ordering by maximal growth rate) is highly comparable (Fig. S2 and S3). The main

difference is that the growth rate on galactose of MG1363 is lower than that of NCDO712 (0.36 h$^{-1}$ ± 0.005 versus 0.54 ± 0.008 h$^{-1}$). As a result, galactose quality is lower than mannitol and similar to ManNAc and gluconic acid for MG1363. This likely results from the absence of the plasmid-encoded tagatose pathway in MG1363 (required for growth on lactose) (37), which can be used in conjunction with the Leloir pathway for galactose consumption in NCDO712 (38).

## Catabolization hierarchy does not strictly follow sugar quality

After establishing the maximum growth rate-based hierarchy of the sugars in the Biolog plate for *L. cremoris* NCDO712, we assessed the sugar catabolization hierarchy in Biolog plates for cells growing exponentially in a bioreactor on glucose, lactose, galactose, and maltose, spanning roughly the observed growth rate range (Fig. 1, left-side for averages and Fig. S4 right-side for replicates). Replicate normalized catabolization rates (from Biolog plates) could be clearly grouped together using Ward-clustering with Euclidian distance, indicating reproducibility (Fig. S4). In Fig. 1, rows and columns were ordered based on the growth rate achieved at a concentration of 0.5% (wt/vol) of the respective sugars (also shown in Fig. S2). The galactose concentration was lower in the reactor from which Biolog plate samples were taken than in the 384-well plate used for determining maximal growth rates (5 mM versus 25 mM), likely resulting in the lower growth rate (μ = 0.44 h$^{-1}$ versus 0.54 h$^{-1}$). The concentration of sugars in the Biolog plate was higher than 25 mM.

The Biolog plate results show that cells were able to catabolize most higher and equal quality Biolog plate sugars (with respect to the sugars cells were growing on), although there are exceptions (Fig. 1; Fig. S4). For example, trehalose catabolization capacity was only detectable in maltose-grown cells, even though trehalose quality is higher than galactose and roughly equal to lactose. With a concentration of 5 mM galactose, cells grow at a rate similar to that of maltose (μ = 0.44 h$^{-1}$ versus 0.45 h$^{-1}$), but they are unable to catabolize maltose. We would have expected this result at higher concentrations of galactose, as the growth rate would then be higher than on maltose. In contrast and in line with a sugar quality-based catabolization hierarchy, under all growth conditions, cells could catabolize the highest quality sugars: glucose, mannose, and N-acetyl-glucosamine (GlcNAc). However, the normalized catabolization rates of GlcNAc were generally lower than those of glucose, in spite of their equal quality. Furthermore, mannose and lactose normalized catabolization rates were higher than that of GlcNAc, even though GlcNAc is of higher quality. The normalized catabolization rate of GlcNAc did increase when cells were cultured on nutrients of decreasing quality. This was also the case for mannose, although the differences in normalized catabolization rates between cells grown on the different sugars were less pronounced. Glucose normalized catabolization rates appeared to always be near their maximum.

In line with a sugar quality-based catabolization hierarchy, galactose- and maltose-grown cells could catabolize the higher-quality sugars: lactose, fructose, galactose, and lactulose. Additionally, the catabolization rates of galactose and fructose increased with decreasing nutrient quality during growth. Furthermore, lactose catabolization rates were always maximal or not detectable in the case of cells grown on glucose. Lactulose (a disaccharide of galactose and fructose) is of similar quality to galactose; however, cells catabolize this sugar more rapidly when grown on galactose than when grown on lower-quality maltose. Lactose-grown cells showed lower but detectable catabolization of lower-quality lactulose with respect to galactose-grown cells.

Glucose-, lactose-, and galactose-grown cells could not catabolize the lower-quality sugars mannitol and gluconic acid. However, these cells could catabolize many other lower-quality sugars to a variable degree (Fig. 1; Fig. S4). For example, glucose- (μ = 0.74 h$^{-1}$) and lactose-grown cells (μ = 0.66 h$^{-1}$) could catabolize lower-quality fructose (μ = 0.57 h$^{-1}$); lactose-grown cells catabolized galactose (μ = 0.54 h$^{-1}$) and lactulose (μ = 0.53 h$^{-1}$); and lactose-, galactose- (μ = 0.44 h$^{-1}$), and maltose-grown cells (μ = 0.45 h$^{-1}$) catabolized the lower-quality sugar N-acetyl-mannosamine (ManNAc, μ = 0.35 h$^{-1}$).

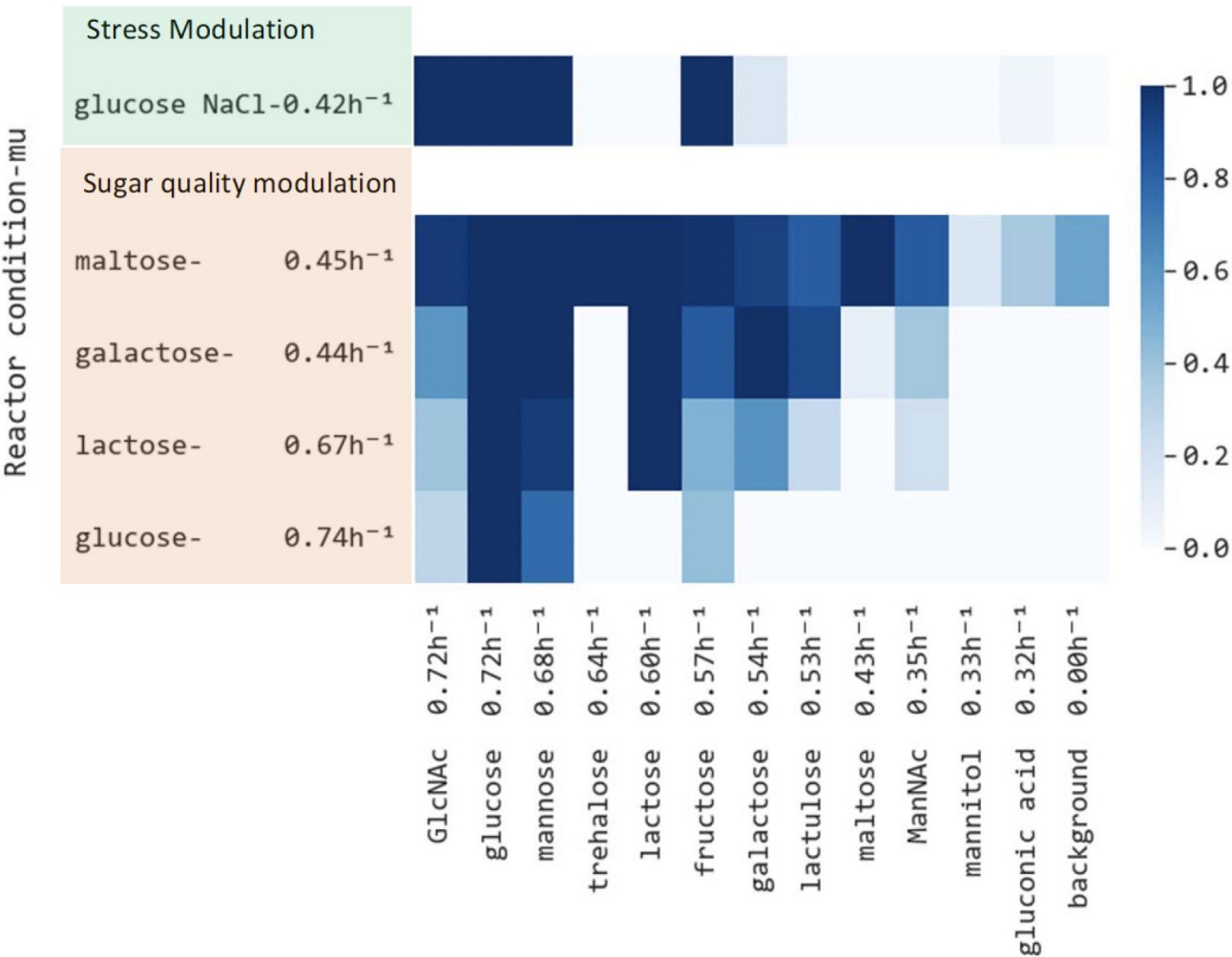

**FIG 1** Heat map of average specific Biolog catabolization rates ($n = 4$, $n = 2$ for lactose due to technical issues) (color legend right-side) normalized to the rate at which the sugar cells had been growing (Bioreactor condition, y-axis). Catabolization rates were determined in the presence of chloramphenicol and erythromycin. Labels on the x-axis include the corresponding maximum achievable growth rate on each sugar as measured in separate microtiter plates (see also Fig. S2). GlcNAc, N-acetyl-glucosamine; ManNAc, N-acetyl-mannosamine.

Similar to fructose, GlcNAc, galactose, and mannose, the catabolization rate of ManNAc did follow the nutrient quality of the different sugars the cells were growing on. We cannot tell if maltose-grown cells could catabolize mannitol and gluconic acid, as this fell within the background noise (which was higher in the case of maltose-grown cells than of other sugars). A comparison of our phenotypic data with transcriptome data from *L. cremoris* MG1363 and its *ccpA* deletion mutant (20) corroborates our findings (Fig. S3).

We conclude that *L. cremoris* catabolization hierarchy does not strictly follow the sugar quality order. There are cases where lower-quality sugars are catabolized (e.g., fructose, galactose, and ManNAc catabolization in the case of lactose-grown cells). Additionally, the higher-quality trehalose is not catabolized during growth on lower-quality galactose, and the GlcNAc catabolization rates are lower than those of glucose, despite their comparable quality. However, we do see that at different growth rates, higher-quality sugars (e.g., glucose, mannose, and GlcNAc) can commonly be catabolized, whereas this is not the case for lower-quality sugars (e.g., mannitol, gluconic acid, ManNAc, and maltose). In addition, the normalized catabolization rates of many sugars often follow the sugar quality during growth (i.e., GlcNAc, mannose, fructose, galactose, and ManNAc).

## Absolute growth rate has only a minor influence on the extent to which sugars can be catabolized

To assess a sugar quality-independent reduction of the growth rate, we imposed osmotic stress (addition of 500 mM NaCL, $\mu = 0.42$ h$^{-1}$). This assessment allows us to test if sugar quality itself or the absolute growth rate controls catabolization capacity. Compared to cells cultured in glucose without osmotic stress, the normalized catabolization rates were slightly elevated for galactose and gluconic acid (Fig. 1; Fig. S4), whereas these rates were clearly elevated for mannose and fructose when salt stress was added. However, the salt stress resulted in lower biomass yields on glucose, resulting in low cell densities at the moment of cell transfer to the Biolog plate (at higher densities, growth would already stagnate). This resulted in lower absolute catabolization rates in the Biolog plate, which makes these rates less reliable. Nevertheless, the sugars that could be catabolized were very similar between glucose and glucose with NaCl stress. This is also reflected in the normalized catabolization rates of glucose + NaCl-grown cells clustering together with cells growing at a much higher rate on glucose instead of clustering together with normalized catabolization rates of cells growing at a similar rate on maltose and galactose (Fig. S4). Therefore, the sugar type and accompanying concentration rather than the absolute growth rate appear to play a larger role in determining which sugars a cell can catabolize.

## Catabolization capacity is not always reflected in the proteome composition

In addition to catabolization rates, we investigated whether the observed substrate catabolizing capacity is reflected in the synthesis of dedicated pathways (Fig. 2). We determined the proteome composition from the four exponentially growing cultures (glucose, lactose, galactose, and maltose) and compared the synthesis of sugar-metabolizing proteins in these cultures (Fig. 3). As expected from the catabolizing capacity (Fig. 1), all cultures synthesized the glucose/mannose transporter complex PTS$^{man}$ (composed of the proteins PtnABD) at similar levels. Pmi (specific to mannose catabolism) and Glk

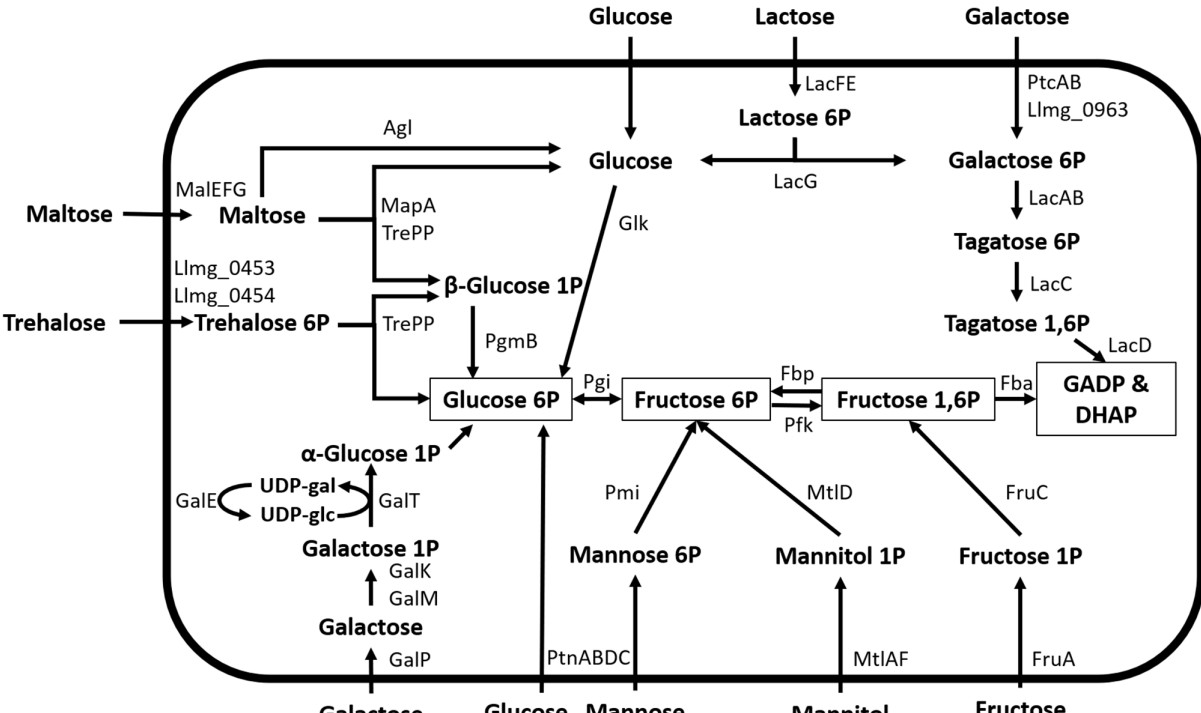

**FIG 2** Overview of metabolic pathways associated with relevant carbon sources. Pathways are shown until Glyceraldehyde 3-phosphate (GADP) and Dihydroxyacetone phosphate (DHAP), after which they all overlap.

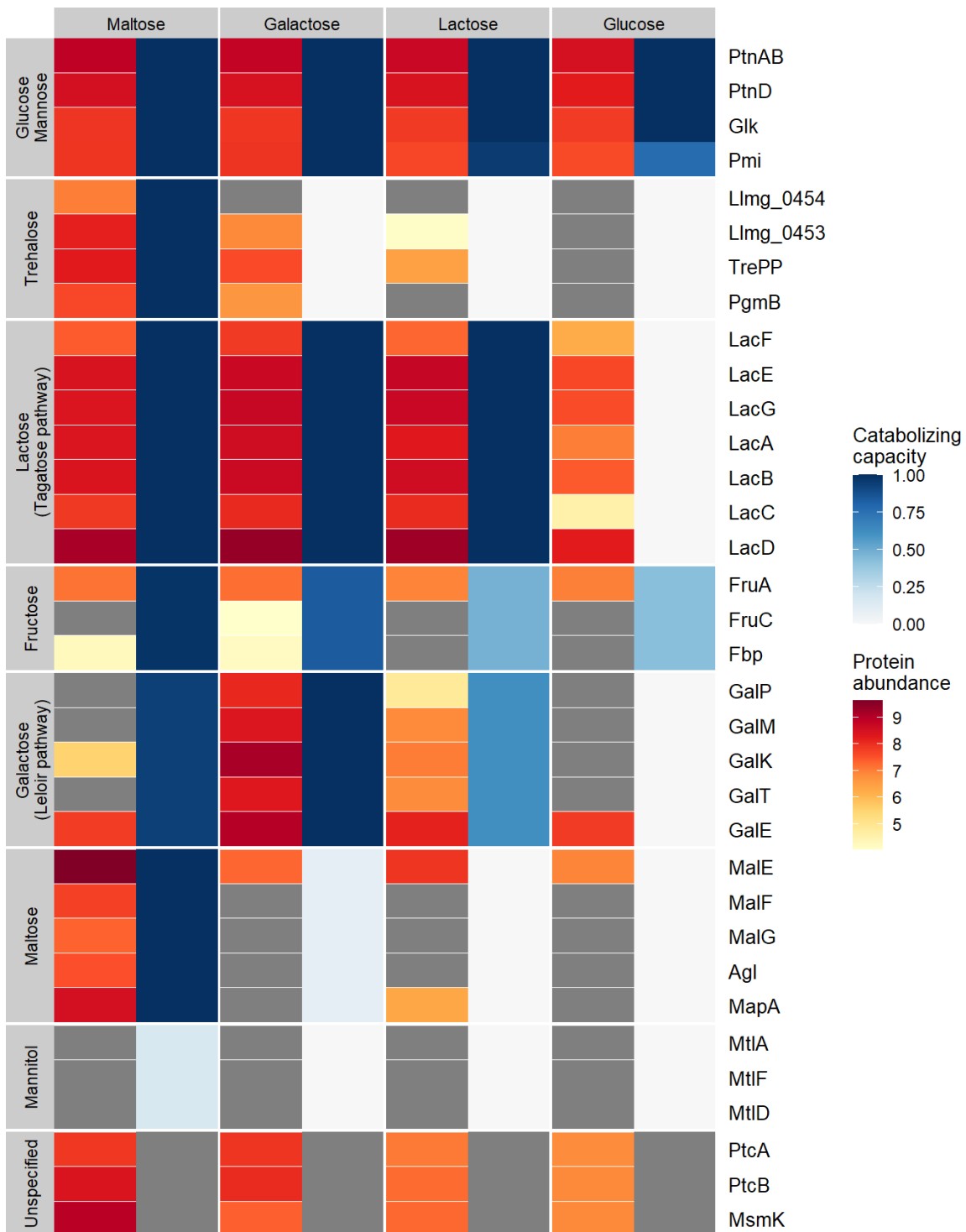

**FIG 3** Average protein abundance (LFQ-based to compare proteins between samples) of proteins involved in sugar metabolism (indicated at the right) after culturing in the four different batch cultures indicated at the top. Vertical groups represent catabolic pathways that are sorted according to the supported growth rate (in descending order) on the indicated sugar. Within each group, proteins are sorted based on consecutive reactions in the pathway. Darker red indicates higher protein levels (log10 scale, shown on the right), while gray indicates that the protein was not detected in at least two of the three replicates. The sugar catabolizing capacity is indicated in the vertical groups given in the blue color scale (shown on the right; no information is given for the unspecified group). Note: PTS$^{man}$ (PtnAB and PtnD) can import both glucose and mannose. Therefore, for PtnAB, PtnD, and Glk, the glucose catabolization capacity is shown, while for Pmi, the mannose catabolization capacity is shown.

(not necessary for mannose catabolism) were also always present at similar levels. This does not align with the observed mannose catabolization rate decrease with increasing sugar quality during growth. Furthermore, the dedicated glucose permease (GlcU) (39) was not detected, while glucose could always be catabolized. This could be due to a lower level of synthesis in combination with the fact that membrane proteins are more difficult to detect with the proteomics method used here (40).

The tagatose pathway proteins, which are associated with utilization of the galactose moiety when growing on lactose, were synthesized in cells growing exponentially on the lower-quality sugars galactose and maltose. These tagatose pathway protein abundances were highly similar to those in lactose-grown cells, which is in line with the observed normalized catabolization rates. Contrary to our expectations, cells growing on glucose also synthesize the tagatose pathway proteins, although at an 18-fold lower level (on average) than lactose-grown cells. This was unexpected considering the higher quality of glucose than of lactose and the lack of lactose catabolization.

Leloir pathway proteins, which are involved in galactose metabolism, were synthesized in both galactose- and lactose-grown cultures, although their abundance was significantly lower in lactose- than in galactose-grown cells. This agrees with the lower rate of galactose catabolization in lactose-grown cells. Importantly, maltose-grown cultures did not synthesize Leloir proteins, even though maltose has a lower quality than galactose. This is in apparent contradiction with the observation that maltose-grown cells were able to catabolize galactose at a higher rate than lactose-grown cells (which did synthesize Leloir proteins). This suggests that maltose-grown cells can catabolize galactose via an alternative pathway, e.g., the tagatose pathway, of which the proteins are highly abundant in maltose-grown cells. However, when catabolized through the tagatose pathway, it is unclear how galactose phosphorylation occurs, which in this pathway normally derives from the imported lactose-phosphate hydrolysis and may involve import through the promiscuous lactose phosphotransferase system (PTS) or the dedicated galactose PTS (38, 41). Notably, two components of this galactose PTS, PtcA and PtcB, were present at significantly higher levels in galactose- and maltose-grown cultures than in lactose- and glucose-grown cultures. This may explain the higher galactose catabolization rate of maltose-grown cells than of lactose-grown cells. However, the third component of the galactose PTS (llmg_0963) (41) was not detected, which may be due to the technical difficulty in the detection of certain membrane proteins even when they are present. Alternatively, PtcA and PtcB may not have been synthesized to prepare for galactose consumption as they are also involved in the transport of other sugars like cellobiose, lactose, and glucose, which requires complex formation with other transmembrane components. However, analogously to the galactose PTS transmembrane component, none of these transmembrane components (i.e., CelB, PtcC, or llmg_1244) were detected (41–44). Finally, proteins for maltose metabolism were not detected in cultures growing on alternative sugars, which is in line with the catabolizing capacity following the sugar quality.

Aside from glucose, lactose, galactose, and maltose pathway proteins, we also examined the presence of pathways required for catabolizing the other sugars in the Biolog plate. While cells could always catabolize fructose to a variable extent, we only detected all fructose-metabolizing enzymes in galactose-grown cells. For maltose-grown cells, we only detected FruA and Fbp, and for lactose- and glucose-grown cells, only FruA was detected. To some extent, this observation agrees with the observation that fructose catabolization rates increase with decreasing nutrient quality during growth, following a nutrient quality-based catabolization hierarchy. Potentially, the abundances of Fbp and FruC were near the detection limit, as their abundances were low even in the case of galactose-grown cells. This may explain why we could not always detect them, despite the detected fructose catabolization.

In line with a nutrient quality-based catabolization hierarchy, proteins related to trehalose metabolism increased in abundance with decreasing nutrient quality. Furthermore, we only detected the complete pathway in maltose-grown cells, which

is in agreement with the lack of trehalose catabolization capacity of lactose-, galactose-, and glucose-grown cells. In galactose- and lactose-grown cells, we did not detect the transmembrane component Llmg_0454, even though this is part of the trehalose transporter complex, which consists of components Llmg_0453 and Llmg_0454. Potentially, Llmg_0454 was not found due to technical limitations in membrane protein detection (40). If that is the case, we would expect to find the catabolization capacity of trehalose in galactose-grown cells, where Llmg_0454 is the only missing component. Potentially, the trehalose catabolization rate was below the detection limit for galactose-grown cells. In glucose- and lactose-grown cells, other trehalose pathway proteins are also missing, which corresponds to the lack of trehalose catabolization in these cells.

Lastly, proteins required for mannitol metabolism, which supports the lowest growth rate, are not detectable in any culture, which is in line with the catabolization hierarchy following the sugar quality order (20, 45). For the remaining carbon sources that could be catabolized by *L. cremoris* (gluconic acid, N-acetyl glucosamine, N-acetyl mannosamine, and lactulose), it is unclear which proteins are involved, hampering their abundance evaluation in the proteome data.

In conclusion, the measured catabolizing capacity does not always reflect protein synthesis. For some sugars, the related catabolization pathways appeared to be synthesized while no catabolization was detected (e.g., trehalose), whereas other sugars could be catabolized while the associated proteins could not be detected (e.g., fructose). Additionally, protein synthesis does not appear to strictly follow a sugar quality-based order, although the synthesis of trehalose pathway proteins decreases with increasing sugar quality. Moreover, proteins for the catabolization of higher-quality sugars are commonly synthesized to higher levels than those required for the catabolization of lower-quality sugars.

## *L. cremoris* synthesizes additional ATP-generating pathways when growing on sugars that support slower growth rates

Synthesizing the proteins related to the catabolization of a given sugar should allow a cell to continue growing without delay after a sudden transition. However, to reach the maximal growth rate of sugar, other proteins must also be adequately synthesized. One set of proteins is related to ATP generation (i.e., glycolysis, pyruvate dissipation, and arginine conversion to ornithine). We investigated the synthesis of proteins associated with these pathways and verified their contribution to cellular metabolism by determining the production level of their metabolic end-products, i.e., the acetate (as a proxy for mixed acid production) to lactate ratio and the specific level of ornithine production (i.e., normalized per $OD_{600}$) using NMR spectroscopy.

The individual glycolytic proteins were present at similar levels in cultures growing on any of the four sugars (glucose, lactose, galactose, and maltose), although the summed abundance of all glycolytic proteins was slightly lower in maltose-grown cultures (Fig. 4A). The pyruvate generated by the glycolysis can either continue into homolactic fermentation using lactate dehydrogenase (Ldh) or into mixed acid fermentation using multiple proteins. Homolactic fermentation is correlated with higher growth rates, while mixed acid fermentation is correlated with lower growth rates. Ldh was consistently synthesized to a high level, irrespective of the sugar used for growth or the corresponding growth rate. In contrast, in cultures growing on the higher-quality sugars glucose and lactose, several proteins for mixed acid fermentation were synthesized to significantly lower levels than in cultures grown on maltose and galactose (Fig. 4A). The elevated level of proteins involved in mixed acid fermentation corresponds with the higher ratio of acetate to lactate (Fig. 5A), although the acetate/lactate ratio is not only controlled by protein levels (46).

In addition, *L. cremoris* NCDO712 can generate additional ATP from arginine consumption via the arginine deiminase pathway. The proteins that convert arginine via citrulline into ornithine and ATP were synthesized to significantly higher levels in the

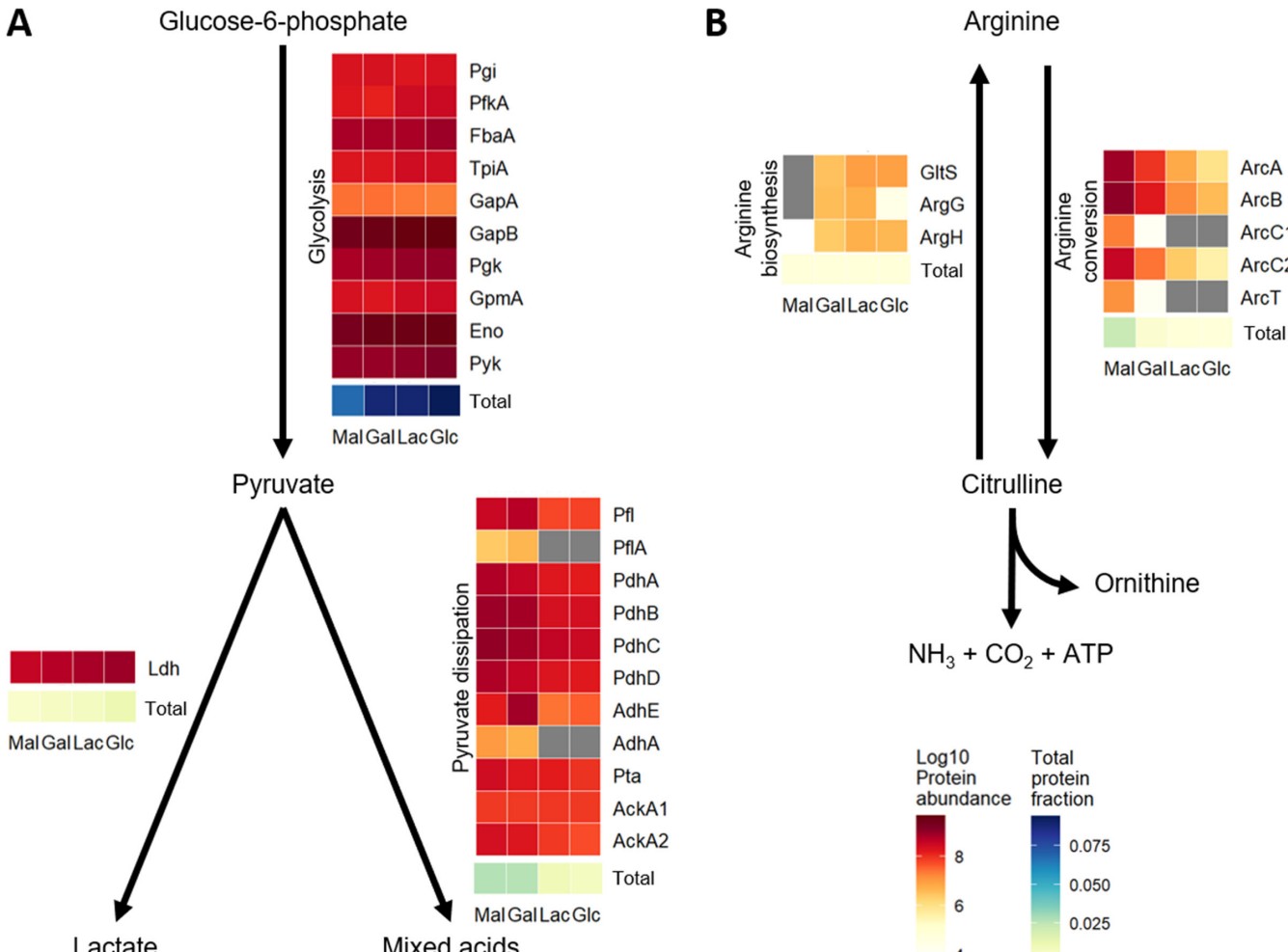

**FIG 4** Average protein abundance (LFQ-based) of ATP-generating pathways in the four different batch conditions (indicated at the bottom of the heat maps). Proteins are sorted according to reaction order. Darker red indicates higher protein abundances, while gray indicates that the protein was not detected (bottom right). The blue-yellow totals indicate the summed amount of all proteins in the pathway (iBAQ-based, bottom right). (A) Glycolysis and pyruvate dissipation. (B) Arginine metabolism.

cultures grown on the lower-quality sugars maltose and galactose than on the higher-quality sugars glucose and lactose (Fig. 4B). Notably, we did not detect the ornithine/arginine antiporter (ArcD1 and ArcD2). Conversely, the arginine and glutamate importer and the proteins involved in the conversion of citrulline into arginine were present at significantly lower levels in maltose-grown cultures. These protein level differences correlated well with the observed higher amounts of ornithine produced by maltose-grown cells than by lactose- and glucos-grown cells, where ornithine production could not be consistently detected (Fig. 5B).

In conclusion, of the different ATP-generating pathways and proteins, the synthesis of glycolysis and Ldh is not influenced by the carbon source or growth rate. In contrast, the proteins that are often associated with lower growth rates (mixed acid fermentation and arginine conversion) (27, 46) are synthesized at lower levels in cultures growing on glucose or lactose (high quality) than in cultures growing on maltose or galactose (low quality). This finding implies that cultures grown on lower-quality sugars are prepared for growth on higher-quality sugars, but not vice versa.

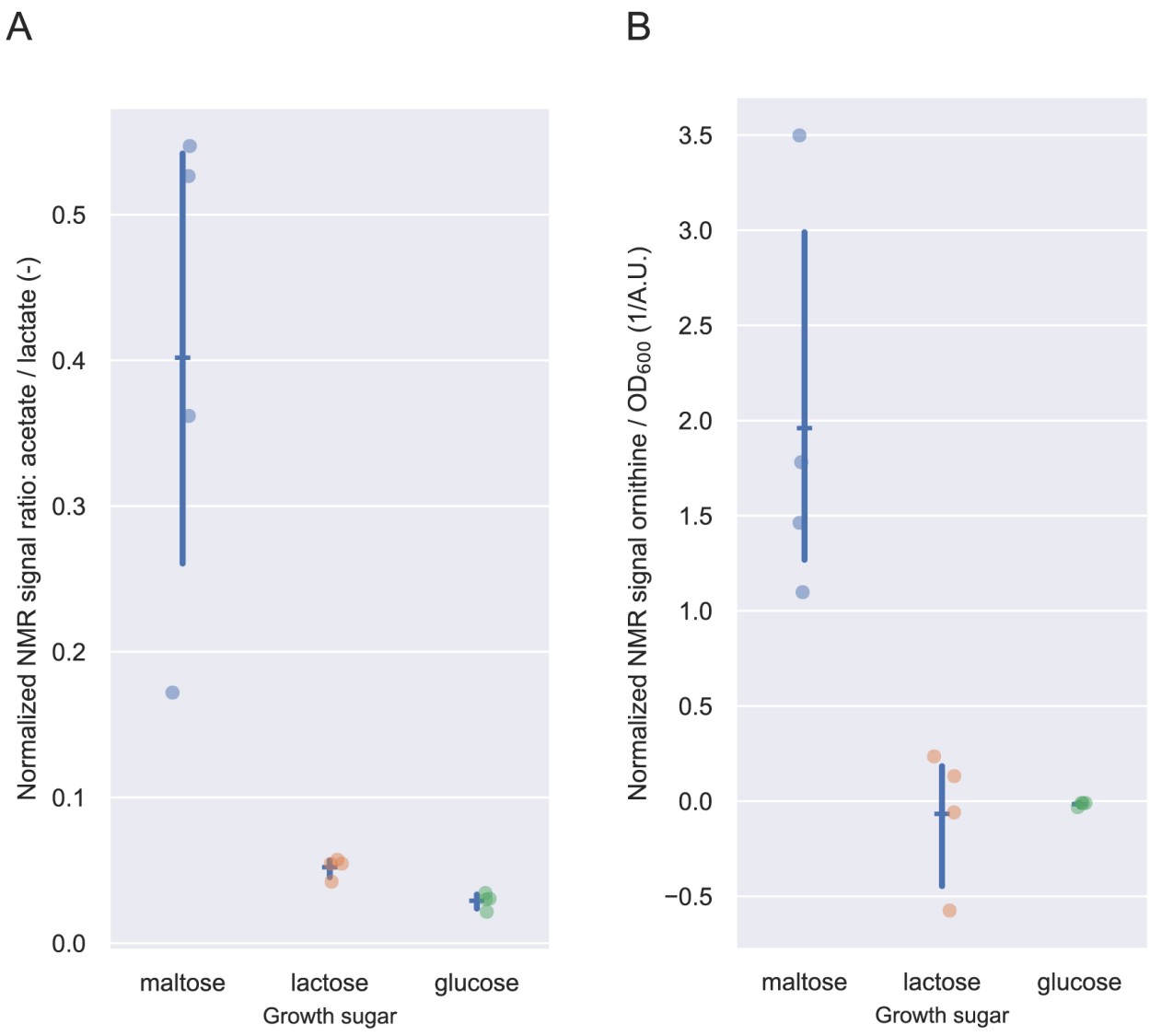

**FIG 5**  (A) Average (horizontal blue line) ratio of normalized NMR signals of acetate over lactate with a 95% confidence interval (vertical blue line); dots indicate replicates ($n = 4$). (B) Average (horizontal blue line) NMR signal of ornithine normalized per $OD_{600}$ in A.U. (absorption units) with a 95% confidence interval (vertical blue line); dots indicate replicates ($n = 4$). Due to technical issues, galactose samples are not available.

## Sugar-specific growth rate alterations influence ribosomal protein synthesis

Besides sugar-specific proteins and general proteins involved in ATP generation, we also investigated how the remaining proteome changed as a function of the sugar-specific growth rate. Therefore, we analyzed which proteome changes are associated with changes in growth rate using redundancy analysis. This analysis revealed that 33% of the total proteome variation was associated with changes in growth rate (Fig. 6A). Subsequent gene set enrichment analysis (Fig. S5) identified that increased growth rates were associated with increased synthesis of translation (i.e., ribosomes and aminoacyl-tRNA) and nucleotide metabolism (i.e., purine and pyrimidine metabolism) proteins and with decreased synthesis of proteins involved in carbohydrate metabolism (i.e., pyruvate dissipation).

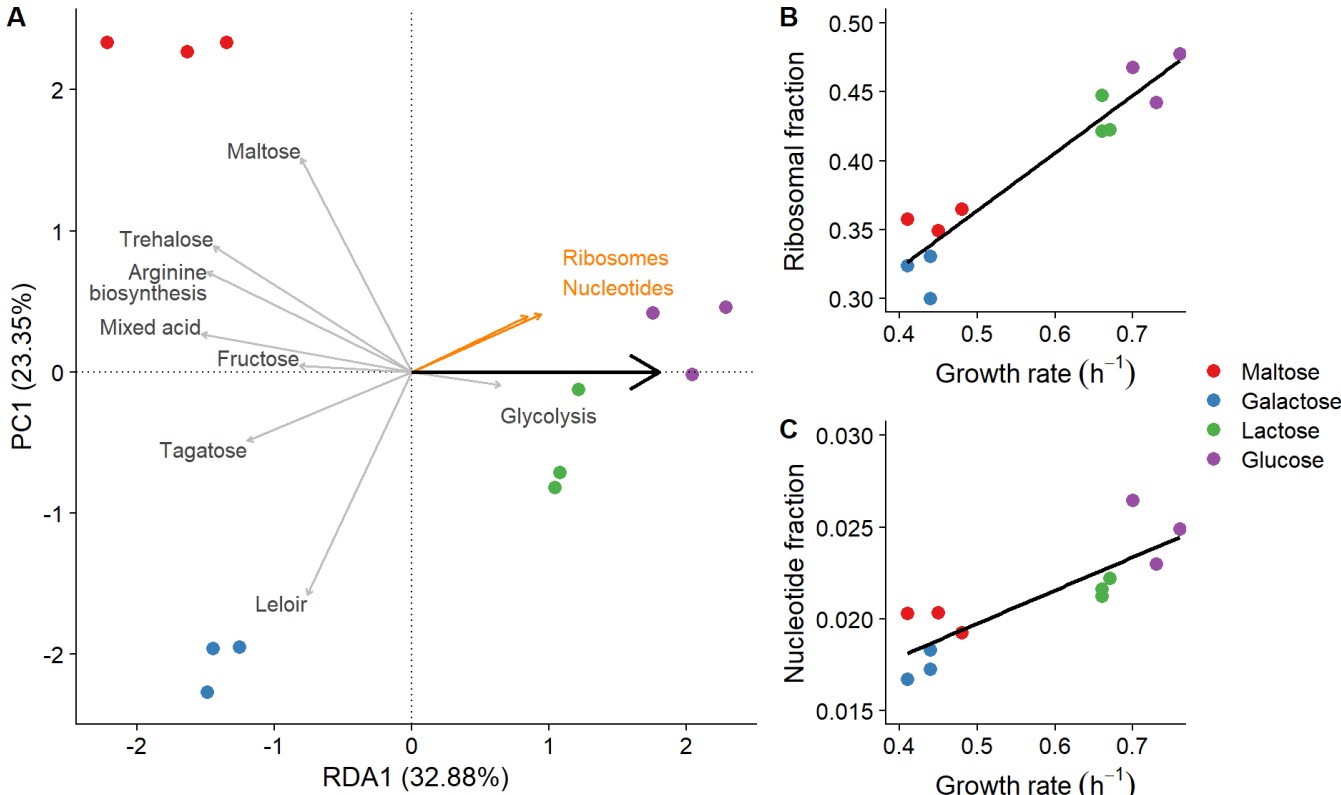

**FIG 6** Proteins that correspond to growth rate. (A) Redundancy analysis constraint on the growth rate (x-component) with the y-component displaying the first principal component. The large arrow indicates the direction of the growth rate increase; the orange arrows indicate the direction of the enriched protein classes; the gray arrows indicate the direction of the protein classes as defined in Fig. 3 and 4 that were discussed in the previous sections. (B) Summed ribosomal protein abundances relative to the total measured proteome (fraction) correlated with the growth rate of the different samples colored by sugar type. (C) Summed fraction of proteins involved in pyrimidine and purine metabolism with the growth rate in the different samples.

The total ribosomal fraction (i.e., the summed amount of all ribosomal proteins) displayed a strong correlation with growth rate (Fig. 6B; Pearson: 0.95; $P$-value: 2E−6), ranging from 36% to 46% (a 1.3-fold increase) of the total proteome in maltose (lowest growth rate) and glucose (highest growth rate) cultures, respectively. Analogously, the proteome fraction involved in purine and pyrimidine metabolism also correlated significantly with growth rate (Fig. 6C; Pearson: 0.86; $P$-value: 4E−4) and ranged from 1.7% to 2.5% (a 1.5-fold increase). No significant correlation was found for the aminoacyl-tRNA fraction.

In conclusion, the ribosomal fraction is the largest fraction of the proteome that increases with increasing growth rates. However, the relative increase of the nucleotide fraction was higher than that of the ribosomal fraction, and hence it is also expected to play an important role in growth rate maximization after a transition. Hence, cultures grown on sugars supporting lower growth rates are not prepared for these characteristics of faster growth.

## The nongrowth-related proteome investment is larger when transitioning to sugars with slower growth rates

The above sections show that when cells transition to a new sugar (i.e., not encountered for at least 20 generations), the proteome must always be adapted. Changes in proteome composition requires both protein production and degradation, but protein degradation is assumed to be mainly achieved by dilution through growth (47–51). We determined the percentage of total proteome that needs to be produced to transition from one sugar to the other (i.e., the proteome investment). To achieve this, we compared the

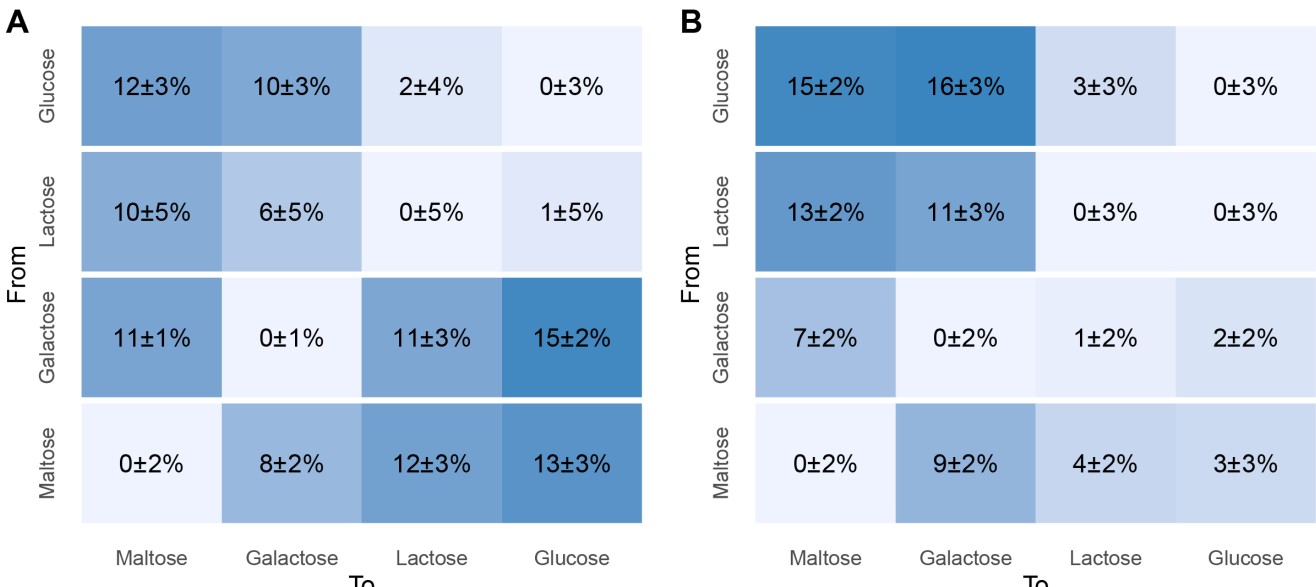

**FIG 7** Protein investment required to fully adapt to another sugar based on (A) all proteins (1,076 proteins) and (B) nongrowth-related proteins only (989 proteins, i.e., no ribosomes and nucleotide metabolism proteins). Values indicate the total fraction of the proteome that needs to be produced, corrected for the variation between different replicates of the same sugar type (diagonal).

proteomes of cells growing for a minimum of 20 generations on two different sugars. We chose 20 generations of growth to avoid carry-over of proteins related to a previous environment. The total proteome investment required for the transition between glucose and lactose, which both support higher growth rates, is relatively limited (≤2%) when compared to transitions that require 6% to 15% of the total proteome to be newly produced (Fig. 7A). The required proteome investment to achieve complete adaptation increases with transitions associated with larger growth rate differences, irrespective of the direction of transition (i.e., to a lower or higher-quality sugar). However, transitions from lower-quality sugars to higher-quality sugars are associated with the production of large amounts of ribosomes (Fig. 6), which likely constitutes a major proportion of the proteome investment in such transitions. Therefore, we assessed the protein investment required during sugar transitions with the exclusion of the ribosomal (and nucleotide metabolism) protein fraction (Fig. 7B). This revealed that the remaining nongrowth-related proteome investment that is required to transition to glucose or lactose from maltose or galactose was relatively limited (≤4%), whereas the inverse transitions required a substantially larger proteome investment (11%–16%; Fig. 7B). These ribosome and nucleotide metabolism-exempt investments can mainly (5% to 9%) contribute to the production of proteins involved in sugar metabolism and additional ATP generation (i.e., pyruvate dissipation and arginine conversion; see also above).

In conclusion, full adaptation upon transitioning from a low to a higher-quality sugar (i.e., to glucose or lactose) requires the investment of mainly growth rate-associated functions (e.g., ribosomal proteins and nucleotide synthesis). This indicates that cultures can likely continue growth without delay on these types of sugars, should they appear, although at an initially lower growth rate. In contrast, transitioning to sugars that support lower growth rates (i.e., maltose or galactose) requires the synthesis of metabolic pathway proteins (i.e., sugar-specific catabolic proteins and proteins related to ATP generation). This is likely to result in a period of growth arrest or lag phase.

## DISCUSSION

A rapid response to transitions between different carbon sources is an important determinant for the fitness of an organism in dynamic natural environments. This is also relevant in industrial fermentations that include mixed substrates or transitions between different media. Therefore, we determined which sugars *L. cremoris* could catabolize while growing at different growth rates on either glucose, lactose, galactose, or maltose.

During growth on lower-quality sugars, *L. cremoris* NCDO712 catabolized a broader range of sugars and expressed a more extensive repertoire of sugar metabolizing pathways, with some exceptions (Fig. 1, Fig. 3). The deviations from the sugar quality hierarchy are the same for *L. cremoris* NCDO712 grown in CDMpc and those based on the transcriptome data of Zomer et al. for *L. cremoris* MG1363 (20). The fact that in *L. cremoris* NCDO712 some trehalose pathway proteins are synthesized during growth on galactose and lactose, and their levels decrease with increasing sugar quality, indicates that sugar quality does play a role in repression for trehalose in NCDO712 (Fig. 3); this cannot be determined for MG1363 from the experiment of Zomer et al., where repression strength was determined by comparing a wild-type and CcpA deletion mutant, both growing on glucose in M17 (20).

NCDO712 can also consume fructose when growing on higher-quality glucose and lactose while not all fructose proteins were detected during growth on maltose, lactose, and glucose. Potentially, the fructose catabolization activity is a result of mannose PTS presence, which is known to be able to transport fructose (52–54).

Despite the deviations from a sugar quality-based repression hierarchy, there is a general tendency to repress poor-quality sugar pathways more than those for high-quality sugars in MG1363, which is comparable to our findings for NCDO712 (20) (Fig. S2). In line with this tendency, MG1363 repressed both mannitol and galactose equally, corresponding to roughly equal sugar quality. In contrast, for NCDO712, mannitol is of lower quality than galactose, and it appears to act accordingly by not showing mannitol catabolization capacity on any of the conditions tested, whereas galactose could be consumed when pregrown on, e.g., maltose. Similar to PTS$^{man}$ (glucose importer) importing fructose, galactose consumption of NCDO712 pregrown on maltose may be a side effect of expressing proteins for lactose consumption, as the lactose PTS has been shown to also import galactose (38). This side effect potentially results in a reduced fitness cost for expressing currently useless proteins, as the sugars can then still be imported (i.e., detected) without expressing the respective sugar catabolizing pathway. This ensures a short lag phase when the respective sugar appears. Enzyme promiscuity like this has also been reported to enhance fitness in situations where an organism must adapt to grow on a previously unencountered carbon source (55–57).

In apparent contradiction with our findings, Qian et al. showed that *L. lactis* strain 65.1 co-consumes maltose and galactose (after preculture on galactose) (17). We find that galactose-grown cells could not catabolize maltose, while maltose-grown cells could catabolize galactose (through the tagatose pathway). Potentially, these cells may first consume galactose and then maltose. The apparent discrepancy between our findings and those of Qian et al. could be strain-specific.

As shown with several examples above, knowledge of the sugar quality order of *L. cremoris* has limited value in predicting which sugars it will be able to catabolize when growing on a specific sugar. Except for trehalose, if a sugar is of higher quality, NCDO712 can catabolize this sugar, but not always at the same rate as sugars of equally high quality. Furthermore, if a sugar is of lower quality, NCDO712 will, in some cases, still be able to catabolize it or express related proteins.

In several cases, NCDO712 synthesizes proteins that do not convey detectable catabolization capacity, even though synthesizing redundant proteins can result in a lower maximal growth rate (58). It is conceivable that the evolutionary history of the strain may explain the observed synthesis of deviations from a systematic hierarchy. This has been shown in yeast, where aside from the quality of different sugars, the frequency at which they encountered certain substrates also influenced gene expression (6).

If a cell can catabolize a certain sugar, this will prevent a period of growth arrest upon transition (6, 18). However, while growth may continue directly, we found that the proteome composition needs to be adjusted to reach the maximal growth rate. This is in line with the strong positive correlation between the ribosome fraction and the growth rate (46, 59). We found that Ldh (homolactic acid fermentation) and glycolytic proteins are present at similar levels regardless of the sugar NCDO712 was growing on, even though the product spectrum of maltose-grown cells was shifted toward mixed acid fermentation relative to glucose- and lactose-grown cells (4, 5). This is similar to the findings by Goel et al. when comparing *L. cremoris* MG1363 at different growth rates in glucose-limited chemostats (46). *E. coli* shows similar behavior, where growth occurred instantly after a nutrient upshift, but it took several hours to reach the maximum growth rate (50).

Lastly, we found that growth rate alone does not always determine which sugars a cell prepares for. Cells growing more slowly on glucose as a result of salt stress (at a rate similar to that on maltose) showed very similar Biolog catabolization profiles to glucose-grown cells without salt stress. We expect glycolytic flux to regulate the catabolization capacity rather than growth (although these are linked in many cases) (1). Potentially, the glycolytic flux was not reduced in these slower-growing salt-stressed cells. However, this would need to be verified, and we cannot rule out that the type of sugar being consumed also influences the catabolization capacity, regardless of flux.

Knowledge about the utilization hierarchy of nonlactose sugars by dairy and nondairy-derived lactococci is relevant when considering the increased demand for plant-based dairy alternatives (60). Such knowledge will allow for the design of starter cultures that achieve fast product acidification and desired flavor profiles (61, 62).

In conclusion, while lactococcal cells in some cases repress and express genes according to a sugar quality order, factors like enzyme promiscuity and evolutionary history may cause cells to deviate from this behavior.

## ACKNOWLEDGMENTS

This study was funded by the Top Institute Food and Nutrition (TIFN, Program 16MF01, Wageningen, The Netherlands).

S.D., B.T., M.K., and H.B. conceived and designed the study. S.D., B.v.O., S.B., Y.L., and X.L. carried out the experiments. All authors analyzed the data. S.D., B.v.O., B.T., M.K., and H.B. wrote the paper.

## AUTHOR AFFILIATIONS

[1]TI Food and Nutrition, Wageningen, the Netherlands
[2]Systems Biology Lab, Vrije Universiteit Amsterdam, Amsterdam, the Netherlands
[3]Host-Microbe Interactomics, Wageningen University & Research, Wageningen, the Netherlands
[4]Laboratory of Biochemistry, Wageningen University & Research, Wageningen, the Netherlands
[5]MAGNEtic resonance research FacilitY (MAGNEFY), Wageningen University & Research, Wageningen, the Netherlands
[6]Microbiology Department, NIZO Food Research, Ede, the Netherlands

## AUTHOR ORCIDs

Sieze Douwenga  http://orcid.org/0000-0003-3604-7524
Berdien van Olst  http://orcid.org/0000-0001-8596-0171
Sjef Boeren  http://orcid.org/0000-0002-0847-8821
Bas Teusink  https://orcid.org/0000-0003-3929-0423
Jacques Vervoort  http://orcid.org/0000-0002-0091-5687
Michiel Kleerebezem  https://orcid.org/0000-0001-8552-2235

Herwig Bachmann  http://orcid.org/0000-0002-8224-0993

## AUTHOR CONTRIBUTIONS

Sieze Douwenga, Conceptualization, Data curation, Formal analysis, Investigation, Methodology, Writing – original draft, Writing – review and editing | Berdien van Olst, Conceptualization, Data curation, Formal analysis, Investigation, Methodology, Visualization, Writing – original draft, Writing – review and editing | Sjef Boeren, Formal analysis, Methodology | Yanzhang Luo, Data curation, Investigation, Methodology | Xin Lai, Investigation, Methodology | Bas Teusink, Conceptualization, Funding acquisition, Supervision, Writing – review and editing | Jacques Vervoort, Formal analysis, Investigation, Methodology, Supervision | Michiel Kleerebezem, Conceptualization, Funding acquisition, Investigation, Supervision, Writing – original draft, Writing – review and editing | Herwig Bachmann, Conceptualization, Funding acquisition, Investigation, Project administration, Resources, Supervision, Writing – review and editing

## ADDITIONAL FILES

The following material is available online.

### Supplemental Material

**Supplemental Material (Spectrum02248-23-S0001.pdf).** This document contains all supplemental information and figures including captions.

### Open Peer Review

**PEER REVIEW HISTORY (review-history.pdf).** An accounting of the reviewer comments and feedback.

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
