## [Reviewer comments · Microbiology Spectrum]

Microbiology Spectrum

The hierarchy of sugar catabolization in *Lactococcus cremoris*

Sieze Douwenga, Berdien van Olst, Sjef Boeren, Yanzhang Luo, Xin Lai, Bas Teusink, Jacques Vervoort, Michiel Kleerebezem, and Herwig Bachmann

Corresponding Author(s): Herwig Bachmann, NIZO Food Research BV

Review Timeline:

Submission Date:	May 29, 2023
Editorial Decision:	July 14, 2023
Revision Received:	September 14, 2023
Accepted:	September 21, 2023

Editor: Angela Re

Reviewer(s): Disclosure of reviewer identity is with reference to reviewer comments included in decision letter(s). The following individuals involved in review of your submission have agreed to reveal their identity: Fabian Moritz Commichau (Reviewer #1); Ok Bin Kim (Reviewer #3)

Transaction Report:

DOI: <https://doi.org/10.1128/spectrum.02248-23>

July 14, 2023

Dr. Herwig Bachmann
NIZO Food Research BV
Kernhemseweg 2
Ede 6718ZB
Netherlands

Re: Spectrum02248-23 (The hierarchy of sugar catabolization in *Lactococcus cremoris*)

Dear Dr. Herwig Bachmann:

Link Not Available

Sincerely,

Angela Re

Journals Department
Reviewer comments:

Reviewer #1 (Comments for the Author):

With interest I was reading the manuscript by Douwenga and Gangwal and co-workers about the hierarchy of sugar catabolism in *Lactococcus cremoris*. I have a few issues that need to be addressed prior to the acceptance of the manuscript. I hope that my report will be helpful in assessing the paper.

Major point

It is unclear to the reader why the authors tested the effect of increased osmolarity on the consumption of carbon sources. Please explain!

Minor points

1. Page 2, line 24: better "synthesis" than "expression"
2. Page 2, line 34: the term "degree of preparation" is a bit unusual. Genes are either expressed or not expressed. Alternatively, some genes are transcribed at a low level.
3. Page 3, lines 45 and 60: better "synthesis" than "expression"
4. Page 4, line 77: "las" operon?
5. Page 4, line 79: better "synthesize" than "express"
6. Page 4, line 102: citation!
7. Page 5, line 124: please provide the number for the reference here
8. Page 5, line 134: "...is based on preparation for higher...", please rephrase the sentence.
9. Page 6, line 147: "12.5mM", very often spacing is missing between the numbers and the units. Please check the whole manuscript.
10. Page 17, line 496: Better "lactate dehydrogenase (Ldh)"
11. Page 19, discussion section. The discussion is rather long. I would recommend to shorten it and to focus on the most interesting points.
12. Page 22, line 666: what is PgmB doing?
13. Page 22, line 669: better "likely needs to be adjusted."
14. Page 26, line 804: please correct the reference. The page numbers are missing. The same is true for the reference 39 (see page 27, line 823).
15. Page 32, line 912: better "pyruvate" than "Pyruvate"

Reviewer #2 (Comments for the Author):

In this manuscript, the authors study the sugar utilization hierarchy in *Lactococcus cremoris*. By combining growth experiments (in bioreactors and multiwell plates) with proteomics data analysis, they investigate the order and preference in which the bacterium makes use of different sugars and related compounds, and describe the utilization of different metabolic pathways. This manuscript presents interesting experiments and is mostly well written. In some cases, the text could be shortened and made more concise. The figures are of good quality, and the methods are described in sufficient detail. The manuscript could be further improved by addressing the following points:

- 1) There does not seem to be a complete map of the metabolic network of *L. cremoris* in the figures or supplementary figures. In my opinion, it is very important to include such a figure and to indicate which metabolic routes are used for the catabolization of the different sugars that are investigated in this study.
- 2) In general, I am surprised that the authors take growth rate as the key parameter to define sugar quality. What about biomass yield as an additional parameter? And wouldn't it be relevant to take the metabolic routes that are required for catabolization of each sugar into account? If sugar A is catabolized via a certain metabolic route, and sugar B is catabolized via (largely) the same metabolic route, it seems plausible that assimilation of sugar A prepares *L. cremoris* for (nearly) immediate assimilation of sugar B. This should be taken into consideration when studying and discussing the utilization hierarchy of sugars.

Minor points:

- L. 27-28: Does this sentence mean that the genes for the catabolization of higher quality sugars are constitutively expressed (also in the absence of these sugars), or are they only expressed in the presence of these higher quality sugars? This should be clarified.
- L. 64: HPr should be spelled out/defined the first time it is mentioned.
- L. 145 ff.: M17 is not defined/explained. 'phase' is missing in L. 146. 'Stationary phase' is mentioned several times. Which OD does this correspond to?
- L. 162/163: Why were different concentrations of these four sugars provided? Wouldn't it make more sense to provide the same concentrations for each of the sugars?
- L. 299/300: the difference between 'absolute growth rate' and 'maximum achievable growth rate' is not completely clear to me.

Reviewer #3 (Comments for the Author):

The manuscript by Douwenga et al. studied the utilization of *L. cremoris* for various sugars through growth experiments and proteomics. First of all, authors found that *L. cremoris* can grow on 12 sugars out of the tested 96 different sugars, and the growth rate was analyzed with these 12 sugars. And they ordered the 12 sugars in hierarchy (high vs. low quality) according to the degree of well supporting growth. And the analysis about proteome with catabolization capacity showed that key enzymes and key transporters for the high quality sugars were abundant consistently, but not for the low quality sugars. Further analysis showed that *L. cremoris* require alternative energy metabolism during growth on the low quality sugars. And finally authors analyzed the proteome investment for sugar transition. Overall, this paper deals with a very important topic and the experiments

and procedures are very well devised. By the way, the manuscript needs to be rewritten more concisely.

Main issues

1. It's hard to get to the point. It takes a lot of concentration to grasp the author's opinion. The authors should rewrite the manuscript concisely and accurately.
For example: Please exclude the explanation about errors during data evaluation for the two sugars cellbiose and maltotriose. You don't have to write it all over again. Correct it in both the main text, abstract, and figures.
 - Line 31: 14 to 12 sugars
 - Line 296 subtitle: 14 to 12
 - Line 305: 14 to 12 different sugars
 - Line 312 ~ Line 318: delete it
 - Line 321: 14 to 12
 - In Graphs of Figure S1, S2, S3, the cellbiose and maltotriose should be deleted
2. Overall, it's hard to figure out what they want to claim. Too much discussion in the result section. Here, the result is the mixture of important results with minor issues.
3. Is the procedure to determine Catabolization capacity appropriate?
4. Figure 2: Catabolizing capacity (Blue color indication) of cells grown on Lactose and Glucose: About Pmi shows different scale. How did you determine the catabolizing capacity related to Pmi?
5. Line 925-926: ~~~protein classes that were discussed in the previous sections: is unclear description in legend.
6. Figure 6 and the related text: Which proteins and how many proteins were analyzed as the "non-growth related proteins (Fig. 6B)"? Authors should clearly define the non-growth related proteins. And how many proteins in total Fig. 6A.
7. In discussion: Avoid repeating the contents, and please write the points concisely.
8. Line 585-586: Inconsistency with the analysis showed in Figure 6.

Minor issues

- Line 24: Transcriptome?
- Line 96, 101, 124...: please check the citing.
- Figure 3A: the total protein fraction in Ldh abundance did not shown

Staff Comments:

Preparing Revision Guidelines

Please return the manuscript within 60 days; if you cannot complete the modification within this time period, please contact me. If you do not wish to modify the manuscript and prefer to submit it to another journal, please notify me of your decision immediately so that the manuscript may be formally withdrawn from consideration by Microbiology Spectrum.

The manuscript by Douwenga et al. studied the utilization of *L. cremoris* for various sugars through growth experiments and proteomics. First of all, authors found that *L. cremoris* can grow on 12 sugars out of the tested 96 different sugars, and the growth rate was analyzed with these 12 sugars. And they ordered the 12 sugars in hierarchy (high vs. low quality) according to the degree of well supporting growth. And the analysis about proteome with catabolization capacity showed that key enzymes and key transporters for the high quality sugars were abundant consistently, but not for the low quality sugars. Further analysis showed that *L. cremoris* require alternative energy metabolism during growth on the low quality sugars. And finally authors analyzed the proteome investment for sugar transition. Overall, this paper deals with a very important topic and the experiments and procedures are very well devised. By the way, the manuscript needs to be rewritten more concisely.

Main issues

1. It's hard to get to the point. It takes a lot of concentration to grasp the author's opinion. The authors should rewrite the manuscript concisely and accurately.

For example: Please exclude the explanation about errors during data evaluation for the two sugars cellbiose and maltotriose. You don't have to write it all over again. Correct it in both the main text, abstract, and figures.

- Line 31: 14 → 12 sugars
 - Line 296 subtitle: 14 → 12
 - Line 305: 14 → 12 different sugars
 - Line 312 ~ Line 318: delete it
 - Line 321: 14 → 12
 - In Graphs of Figure S1, S2, S3, the cellbiose and maltotriose should be deleted
2. Overall, it's hard to figure out what they want to claim. Too much discussion in the result section. Here, the result is the mixture of important results with minor issues.
 3. Is the procedure to determine Catabolization capacity appropriate?
 4. Figure 2: Catabolizing capacity (Blue color indication) of cells grown on Lactose and Glucose: About Pmi shows different scale. How did you determine the catabolizing capacity related to Pmi?
 5. Line 925-926: ~protein classes that were discussed in the previous sections: is unclear description in legend.
 6. Figure 6 and the related text: Which proteins and how many proteins were analyzed as the "non-growth related proteins (Fig. 6B)"? Authors should clearly define the non-growth related proteins. And how many proteins in total Fig. 6A.
 7. In discussion: Avoid repeating the contents, and please write the points concisely.
 8. Line 585-586: Inconsistency with the analysis showed in Figure 6.

Minor issues

- Line 24: Transcriptome?
- Line 96, 101, 124...: please check the citing.
- Figure 3A: the total protein fraction in Ldh abundance did not shown

We would like to thank the reviewers for their valuable feedback on our manuscript. We have implemented most of the suggestions in the revised version of the manuscript and below you can find a detailed response to the individual comments (in blue). As requested, next to others we added a figure to the revised manuscript and rigorously shortened the discussion.

Reviewer comments:

Reviewer #1 (Comments for the Author):

With interest I was reading the manuscript by Douwenga and Gangwal and co-workers about the hierarchy of sugar catabolism in *Lactococcus cremoris*. I have a few issues that need to be addressed prior to the acceptance of the manuscript. I hope that my report will be helpful in assessing the paper.

Major point

It is unclear to the reader why the authors tested the effect of increased osmolarity on the consumption of carbon sources. Please explain!

AU: We added an explanation for this in lines 419-423.

Minor points

1. Page 2, line 24: better "synthesis" than "expression"

AU: We changed the wording throughout manuscript as suggested

2. Page 2, line 34: the term "degree of preparation" is a bit unusual. Genes are either expressed or not expressed. Alternatively, some genes are transcribed at a low level.

AU: we changed wording as suggested

3. Page 3, lines 45 and 60: better "synthesis" than "expression"

4. Page 4, line 77: "las" operon?

5. Page 4, line 79: better "synthesize" than "express"

6. Page 4, line 102: citation!

7. Page 5, line 124: please provide the number for the reference here

8. Page 5, line 134: "...is based on preparation for higher...", please rephrase the sentence.

9. Page 6, line 147: "12.5mM", very often spacing is missing between the numbers and the units. Please check the whole manuscript.

10. Page 17, line 496: Better "lactate dehydrogenase (Ldh)"

11. Page 19, discussion section. The discussion is rather long. I would recommend to shorten it and to focus on the most interesting points.

AU: for all points above we adjusted the manuscript (throughout the text if applicable) as suggested

12. Page 22, line 666: what is PgmB doing?

AU: This point was removed from the discussion while shortening it (in line with comments from reviewer 2 and 3).

13. Page 22, line 669: better "likely needs to be adjusted."

14. Page 26, line 804: please correct the reference. The page numbers are missing. The same is true for the reference 39 (see page 27, line 823).

15. Page 32, line 912: better "pyruvate" than "Pyruvate"

AU: for all points above we adjusted the manuscript (throughout the text if applicable) as suggested

Reviewer #2 (Comments for the Author):

In this manuscript, the authors study the sugar utilization hierarchy in *Lactococcus cremoris*. By combining growth experiments (in bioreactors and multiwell plates) with proteomics data analysis, they investigate the order and preference in which the bacterium makes use of different sugars and related compounds, and describe the utilization of different metabolic pathways

This manuscript presents interesting experiments and is mostly well written. In some cases, the text could be shortened and made more concise.

AU: We addressed this by shortening the results but especially the discussion (in line with suggestions from reviewer 3).

The figures are of good quality, and the methods are described in sufficient detail. The manuscript could be further improved by addressing the following points:

1) There does not seem to be a complete map of the metabolic network of *L. cremoris* in the figures or supplementary figures. In my opinion, it is very important to include such a figure and to indicate which metabolic routes are used for the catabolization of the different sugars that are investigated in this study.

AU: As suggested a map was added to the revised manuscript (figure 2) and it is referenced in the text (line 438 in marked up document)

2) In general, I am surprised that the authors take growth rate as the key parameter to define sugar quality. What about biomass yield as an additional parameter?

AU: The reason for selecting growth rate rather than yield as the defining parameter for sugar quality stems from the rationale that faster growing cells with a lower yield will outcompete cells with a lower growth rate and higher yield in an environment where resources are shared (evolutionary selection pressure in suspension is predominantly on growth rate). See:

Bachmann, et al. "Availability of public goods shapes the evolution of competing metabolic strategies." *Proceedings of the National Academy of Sciences* 110.35 (2013): 14302-14307.

In addition, the idea of resource allocation and metabolic shifts are often related to growth rate e.g., see our reasoning in line 710-711 and references 47 and 62. In *L. cremoris* growth rate is inversely correlated to biomass yield – see Bachmann et al. reference above - which allows to extrapolate the

results, but that should be done with some care as yield measurements are typically noisier than rate measurements.

We added an explanation relating to yield in lines 132 to 135.

And wouldn't it be relevant to take the metabolic routes that are required for catabolization of each sugar into account? If sugar A is catabolized via a certain metabolic route, and sugar B is catabolized via (largely) the same metabolic route, it seems plausible that assimilation of sugar A prepares *L. cremoris* for (nearly) immediate assimilation of sugar B. This should be taken into consideration when studying and discussing the utilization hierarchy of sugars.

AU: We agree for the situations where metabolic routes for different sugars completely overlap or if the difference in proteins is made up by promiscuous enzymatic activity (usually for transporters). If only one enzyme/transporter differs, the preparedness of a cell can be severely impacted. We discuss this for sugars like lactose and galactose, and mannose, glucose and fructose. And we look at this in a more holistic way in figure 7, taking all proteins into account not just the pathway proteins. This is expected to influence the lag phase, but not whether a sugar can directly be catabolized. If there is no complete overlap and no enzyme promiscuity, no catabolic activity for a secondary sugar should be present, even if all but one of the essential enzymes of the secondary sugar are present.

We added a pathway overview to the manuscript (figure 2), to make comparison of the different pathways easier.

Minor points:

L. 27-28: Does this sentence mean that the genes for the catabolization of higher quality sugars are constitutively expressed (also in the absence of these sugars), or are they only expressed in the presence of these higher quality sugars? This should be clarified.

AU: this is a relative sugar quality and therefore, it is not constitutive. We adapted the sentence to clarify this in the revised manuscript.

L. 64: HPr should be spelled out/defined the first time it is mentioned.

AU: adjusted as suggested

L. 145 ff.: M17 is not defined/explained 'phase' is missing in L. 146. 'Stationary phase' is mentioned several times. Which OD does this correspond to?

AU: the supplier of M17 broth and the OD values for sampling were added to the revised manuscript

L. 162/163: Why were different concentrations of these four sugars provided? Wouldn't it make more sense to provide the same concentrations for each of the sugars?

AU: The molarities differ because we aimed for identical w/v percentages. Therefore, the molarity of a disaccharide is half of a monosaccharide. In any case, 12.5mM disaccharide and 25mM monosaccharide are excess conditions for *L. cremoris* meaning that in this range the concentration no longer impacts the growth rate. Due to technical issues the galactose concentration deviates, which causes slower growth rates. This is explained in line 367-370.

L. 299/300: the difference between 'absolute growth rate' and 'maximum achievable growth rate' is not completely clear to me.

AU: as suggested we explain this in more detail in the revised manuscript (line 320-325)

Reviewer #3 (Comments for the Author):

The manuscript by Douwenga et al. studied the utilization of *L. cremoris* for various sugars through growth experiments and proteomics. First of all, authors found that *L. cremoris* can grow on 12 sugars out of the tested 96 different sugars, and the growth rate was analyzed with these 12 sugars. And they ordered the 12 sugars in hierarchy (high vs. low quality) according to the degree of well supporting growth. And the analysis about proteome with catabolization capacity showed that key enzymes and key transporters for the high quality sugars were abundant consistently, but not for the low quality sugars. Further analysis showed that *L. cremoris* require alternative energy metabolism during growth on the low quality sugars. And finally authors analyzed the proteome investment for sugar transition. Overall, this paper deals with a very important topic and the experiments and procedures are very well devised. By the way, the manuscript needs to be rewritten more concisely.

Main issues

1. It's hard to get to the point. It takes a lot of concentration to grasp the author's opinion. The authors should rewrite the manuscript concisely and accurately.

For example: Please exclude the explanation about errors during data evaluation for the two sugars cellobiose and maltotriose. You don't have to write it all over again. Correct it in both the main text, abstract, and figures.

- Line 31: 14 to 12 sugars

- Line 296 subtitle: 14 to 12

- Line 305: 14 to 12 different sugars

- Line 312 ~ Line 318: delete it

- Line 321: 14 to 12

- In Graphs of Figure S1, S2, S3, the cellobiose and maltotriose should be deleted

AU: To maintain completeness of the Biolog dataset we did not remove the cellobiose and maltotriose data from the supplementary figures. However, to increase readability it was moved from the main text to the supplementary information.

2. Overall, it's hard to figure out what they want to claim. Too much discussion in the result section. Here, the result is the mixture of important results with minor issues.

AU: As suggested we shortened both the results and discussion section to improve readability. Especially the discussion section was shortened by removing overlap with the results rigorously. We think this improved readability considerably.

3. Is the procedure to determine Catabolization capacity appropriate?

AU: This method was used earlier for this purpose e.g., Egli et al., Price et al. and Tachon et al. These authors describe the method in detail and they are cited in the paper.

4. Figure 2: Catabolizing capacity (Blue color indication) of cells grown on Lactose and Glucose: About Pmi shows different scale. How did you determine the catabolizing capacity related to Pmi?

AU: We explain this in the last sentence of the figure caption: *Note: PTSman (PtnAB and PtnD) can import both glucose and mannose. Therefore, for PtnAB, PtnD and Glk the glucose catabolization capacity is shown while for Pmi the mannose catabolization capacity is shown.*

5. Line 925-926: ~~protein classes that were discussed in the previous sections: is unclear description in legend.

AU: We intended to refer to the protein classes that can be specifically seen in Figure 3 and 4. In the revised manuscript we changed the legend of figure 6 to include this information.

6. Figure 6 and the related text: Which proteins and how many proteins were analysed as the "non-growth-related proteins (Fig. 6B)"? Authors should clearly define the non-growth related proteins. And how many proteins in total Fig. 6A.

AU: *Note: figure 6 has become figure 7 in the revised manuscript*

AU: The non-growth-related proteins are defined as all proteins identified in the analysis (1076 proteins; Figure 7A) excluding the proteins related to growth (ribosomal proteins and proteins involved in nucleotide metabolism; 87 proteins) and therefore the non-growth-related proteins are the remaining 989 proteins (see also lines 592-595 in the main text). We updated the figure legend to indicate the number of proteins in the analysis. A list of all identified proteins per sample was deposited together with the raw data at the PRIDE repository.

7. In discussion: Avoid repeating the contents, and please write the points concisely.

AU: We rigorously rewrote the discussion as suggested.

8. Line 585-586: Inconsistency with the analysis showed in Figure 6.

AU: *Note: figure 6 has become figure 7 in the revised manuscript*

AU: Please note that the comparison cannot strictly be made. Figure 6a and 6b focus on more than just metabolic proteins. However, if a comparison is made this should be done with figure 6b which focuses on the relevant metabolic routes and other non-growth-related proteins. In that case there is no inconsistency.

Minor issues

- Line 24: Transcriptome?

AU: We adjusted the text to clarify this point.

Line 96, 101, 124...: please check the citing.

AU: Corrected citations.

- Figure 3A: the total protein fraction in Ldh abundance did not shown

AU: The figure was updated to include the total protein fraction of Ldh.

Non-reviewer-based adjustment:

Corrected x-axis of figure S5 (axis previously cut-off on right-side).

September 21, 2023

Dr. Herwig Bachmann
NIZO Food Research BV
Kernhemseweg 2
Ede 6718ZB
Netherlands

Re: Spectrum02248-23R1 (The hierarchy of sugar catabolization in *Lactococcus cremoris*)

Dear Dr. Herwig Bachmann:

Your manuscript has been accepted, and I am forwarding it to the ASM Journals Department for publication. You will be notified when your proofs are ready to be viewed.

Sincerely,

Angela Re
Editor, Microbiology Spectrum
